# Laser Prophylaxis for Dry Age-Related Macular Degeneration: Current Evidence

**DOI:** 10.3390/jcm14124157

**Published:** 2025-06-11

**Authors:** Jeffrey K. Luttrull, Stephen H. Sinclair, Igor Kozak

**Affiliations:** 1Ventura County Retina Vitreous Medical Group, 3160 Telegraph Rd., Suite 230, Ventura, CA 93003, USA; 2LIGHT: The International Retinal Laser Society, 10580 Wilshire Blvd. #88, Los Angeles, CA 90024, USA; stephenhsinclair@icloud.com (S.H.S.); kozak@arizona.edu (I.K.); 3Pennsylvania College of Optometry, Drexel University, Philadelphia, PA 19027, USA; 4Department of Ophthalmology, School of Medicine, University of Arizona, Tucson, AZ 85711, USA

**Keywords:** age-related macular degeneration, neovascularization, laser, prevention, propensity score, real-world data

## Abstract

Purpose: To review the role of prophylactic panmacular laser treatment for age-related macular degeneration (AMD). Method: A narrative review of studies employing laser treatment for non-exudative (“dry”) AMD listed in the PubMed database. Results: In multiple published studies, macular laser treatment that causes laser-induced retinal damage (LIRD) has shown either no overall benefit or has accelerated disease progression and vision loss in eyes with dry AMD, particularly in high-risk eyes with more severe pre-treatment disease. Conversely, other studies, including randomized and matched propensity-scored real-world-data (RWD) clinical studies, indicate that avoidance of LIRD may allow laser to be used safely and effectively to prevent vision loss from AMD by slowing disease progression and preventing neovascular conversion without adverse treatment effects, benefiting the highest-risk eyes most. Conclusions: AMD is the leading cause of irreversible vision loss worldwide. Current evidence suggests that laser prophylaxis for dry AMD may hold great promise in preventing age-related vision loss. More studies, especially prospective clinical trials, from more investigators, are needed. If current evidence is confirmed, laser prophylaxis would rise to a preeminent position as the safest, most effective, and most economical and accessible approach to reducing vision loss from AMD.

## 1. Introduction

Neovascular AMD (NAMD) is a leading cause of irreversible vision loss worldwide, growing in prevalence as the world population ages [1]. Preventing conversion from non-exudative or “dry” AMD to exudative, or “wet”, neovascular AMD should thus take a position of high priority in the minds of ophthalmologists.

As a chronic progressive retinopathy (CPR), AMD is by definition a neurodegeneration and thus part of the predominant class of disease affecting the aging population worldwide [1,2,3,4,5,6]. The most important predispositions to AMD, besides age, include race, nutrition, lifestyle habits such as smoking, and genetic profiles. Like all diseases of aging, the course of AMD is characterized by cumulative cellular dysfunction leading to anatomic disruption and degeneration, fomenting chronic inflammation [3,6]. In the advanced end stages of AMD, vision loss may ensue due to the development of macular geographic chorioretinal atrophy (GA) or NAMD. Historically, the principal therapy for dry AMD has been the use of antioxidant vitamins and lifestyle adjustments such as improved diet and smoking avoidance, whose success can best be described as modest [1,2,3,4,5,6,7,8,9,10].

Retinal laser treatment, like medicine in general, has evolved significantly since it was first introduced into clinical practice over 60 years ago [2]. Originally conceived of and employed as a surgical procedure for local thermal cautery of the retina by photocoagulation, with limited indications, inherent and significant treatment limitations, and adverse treatment effects, retinal laser treatment as currently conceived can now be employed as an effective, broadly applicable, infinitely repeatable and thus sustainable medical therapy without adverse effects, characteristics arguably essential for the prevention of vision loss from neurodegenerative chronic progressive retinopathies (CPRs) that prominently include AMD [3].

Dry AMD is one of the oldest indications for macular laser treatment [2,3,4,5]. However, until only recently, this long history of laser prophylaxis for AMD was marked by disappointment [3,5,6]. As a result, many have abandoned the idea and dismissed the potential of macular laser treatment for the safe and effective preventive treatment of dry AMD [7]. Our review of the literature concerning laser prophylaxis in dry AMD suggests, however, that the failure of treatment may be largely attributed to a single factor which can be easily avoided—laser-induced retinal damage (LIRD)—and that avoidance of LIRD may substantially shift the calculus for success in favor of laser treatment.

## 2. Method

A literature review based on studies of macular laser treatment for dry AMD listed in the PubMed database (http://pubmed.ncbi.nlm.nih.gov/ accessed on 10 January 2025) was conducted. Search terms included macula, laser, age-related macular degeneration, subthreshold, prevention, and prophylaxis. Primary studies reporting clinical outcomes of macular laser for dry AMD were included, except in the case of the prior “laser for drusen” studies, where the majority of studies recommended the use of meta-analyses, such as the Cochrane Database [5]. Secondary and sub-analyses, and reviews were not included. A total of 173 items were returned in PubMed based on the keyword search. Despite the query specifying age-related macular degeneration, the majority of search returns described laser treatment of diabetic macular edema. Excluding the “laser for drusen” studies included in the 2015 Cochrane Database meta-analysis, 9 subsequent primary studies reporting the clinical outcomes of prophylactic macular laser for dry AMD were identified for inclusion in this report [5].

## 3. Results/Discussion

### 3.1. Laser for Cautery: The Damage Hypothesis Applied to Dry AMD

While the use of intense, focused light for various purposes including the treatment of the eye can be traced back a millennium or more, interest in focused-light photocoagulation of the retina began in earnest over 300 years ago [2]. Thus, by the time the laser arrived in the late 1950s, the principle of using intense light for retinal photocoagulation (RPC) was firmly established [2,3,4,8]. What laser added was higher precision and greater control compared to prior methods of intense light delivery [2]. Because devices administering light, and then laser, had only ever been used to burn and destroy the retina as a surgical procedure, there is no record that thought was given to any other application or mode of retinal laser operation [8]. LIRD thus was universally presumed to be both a necessary and sufficiently effective therapy [2,3,4,6,7,8].

Application of RPC to the treatment of dry AMD is illustrative. Macular drusen are considered the sine qua non of dry AMD. The size, location, and number of macular drusen were strongly associated with the primary risks of progression with vision loss from geographic atrophy (GA) and/or neovascular, or “wet”, AMD (NAMD) [9,10,11]. It seemed thus obvious that elimination of macular drusen would be beneficial and even a necessary precondition of improving the prognosis of AMD [2,3,4,5,6,7,8,9,10,11]. In this context, it was noted that drusen adjacent to RPC lesions tended to resolve and disappear, while directly treated drusen tended to resolve as well [4,5,11]. Early results of “laser for drusen” treatment showed promise, showing resolution of drusen following both direct and indirect RPD [3,5,6,7,11,12] (Figure 1). However, with time, subsequent studies found that the efficiency of RPC laser drusen reduction paralleled an increased risk of vision loss, mainly from incitement of NAMD [3,5,6,12]. This is because drusen elimination depended upon the inflammatory response to LIRD. The more intense the damage, the more effective at drusen reduction and the more inflammation and structural compromise inflicted on the RPE–Bruch’s membrane complex [4,5,12]. By compromising this critical barrier to choroidal neovascularization, the LIRD required to induce resolution of drusen via inflammatory debridement also increased the likelihood of NAMD development and visual loss [3,5,6,12]. Thus, the “laser-for-drusen” studies suggested two key principles of macular laser treatment for dry AMD: (1) drusen elimination is not necessarily beneficial; (2) treatment should avoid LIRD and incitement of inflammation [3,5,6,12] (Figure 2).

At the same time that the “laser-for-drusen” studies were showing a lack of treatment benefits, four new retinal laser modalities were appearing [14,15,16,17,18,19]. Despite the failure of laser for drusen, universal agreement persisted on LIRD as the necessary prerequisite to achieve a therapeutic laser effect for all retinal laser indications, including AMD [3,6,7,8,14,15,16,17,18,19]. Thus, these new laser modalities were conceived and developed to maintain, but reduce, LIRD in different ways in the hopes of achieving better results despite doing essentially the same thing for the same reasons [6,8,14,15,16,17,18,19]. By reducing retinal damage compared to conventional RPC, these modes were described as “subthreshold” treatment, because the retinal burns were less visually obvious by biomicroscopy than the classic, white, full-thickness suprathreshold RPC burns employed in the Early Treatment of Diabetic Retinopathy Study (ETDRS) and Macular Photocoagulation Study Group (MPSG) reports [3,6,8,20].

Three “short-pulse” laser platforms were introduced employing similar means and effects, differing slightly by the duration of the laser beam exposure, from nano- to microseconds in length [14,15,16,17,18,19]. The Ellex “2RT” nanosecond laser (Lumibird, Lannion, France) is a photodisruptive YAG laser that uses a 3-nanosecond exposure to vaporize the RPE while minimizing collateral damage to adjacent structures [15,16]. “SRT” laser (for selective retinal therapy, produced by Leutronic (Billerica, Massachusetts)) is another short-pulse laser (1.7-microsecond exposure) designed to similarly limit collateral damage to photoreceptors and Bruch’s membrane by creating thermal cavitation bubbles at the melanosomes of the RPE that kill the cell via internal explosions [17,18]. The explosive nature of the 2RT and SRT lasers results from the brevity of the laser pulse precluding heat dissipation from the point of absorption (the RPE melanosome). The Pattern Scanning Laser (PASCAL) (Iridex, Mountain View, CA, USA), also with a 532 nm wavelength, uses a slightly longer 10–12 us pulse, which allows a small degree of heat dissipation to reduce, but not eliminate, the explosive effect of exposure [19]. The PASCAL, rather than preferentially targeting the RPE like 2RT and SRF, was designed in the 1990s with the hope of treating diabetic retinopathy (DR) by reducing retinal metabolic demand in the retina by attempting to selectively destroy outer segments of retinal photoreceptors [19]. Interestingly, each of these inherently “destructive-by-design” laser modes is described by the makers as “non-damaging” [14,15,16,17,18,19]. Finally, the micropulsed laser (MPL), originally using a near-infrared 810 nm wavelength and later visible wavelengths, could deliver laser energy in trains of 40–100 ms pulses, long enough to allow sufficient heat dissipation from the melanosome at the time of exposure to prevent RPE vaporization or internal cavitation [3,6,21,22,23,24,25,26]. The MPL could be used to heat the RPE without killing it by using appropriate laser parameters [3]. Unlike the short-pulse laser modes, MPL can produce a wide range of tissue effects from sublethal thermal stimulation of the RPE to classic, full-thickness suprathreshold ETDRS-style RPC [21]. As noted, the original MPL units employed 810 nm wavelengths poorly absorbed by the RPE, making the creation of RPC burns with 810 nm more difficult than with higher energy and more intensely absorbed visible wavelength lasers such as yellow (577 μm) and green (532 nm) [3,6,21,22,23,24,25,26]. Thus, early MPL users intending to cause LIRD employed pulse frequencies of usually 15% or more to approximate the thermal behavior of a CW laser to enhance the creation of LIRD, although generally at a reduced level of severity compared to traditional suprathreshold CW RPC [3,4,5,6,8,27].

Like the laser-for-drusen studies employing CW RPC, these new approaches, employing less severe LIRD in dry AMD, were also unsuccessful [14,15,16,17,18,19]. In 2010, a randomized clinical trial (RCT) using the SRT laser to attempt to slow the progression of age-related geographic atrophy (GA) by treating the margins of the GA lesions was discontinued prior to completion due to rapid doubling of the rate of GA progression [17,18]. Further echoing the “laser for drusen” studies, Querques and associates noted a decrease in reticular pseudodrusen (RPD), a major risk factor for NAMD three months following short-pulse PASCAL CW laser treatment [19]. However, no other treatment benefits were reported, nor longer-term follow-up. More recently, an RCT of nanosecond 2RT laser photodisruption of the RPE demonstrated reduced drusen density but failed to otherwise improve early AMD, while accelerating disease progression and vision loss in eyes with higher-risk dry AMD, particularly eyes with reticular pseudodrusen (RPD) and pre-existing GA, prompting warnings against any type of laser treatment for dry AMD [9,15,16].

LIRD: Sufficient, but necessary?

As noted above, MPL technology can produce a wide range of treatment effects depending upon the treatment parameters [3,6,21,23,24]. While 810 nm MPL at a 10% or higher DC may produce suprathreshold RPC, LIRD at a 5% DC has never been reported, illustrating the important influence of pulse frequency [3,23]. “SDM^TM^” for “low-intensity/high-density subthreshold diode micropulse laser” was introduced in 2000 [22]. SDM described a specific application of MPL employing a long wavelength (810 nm) and low (5%) DC to perform treatment that was reliably sublethal to the RPE, thus without LIRD or adverse treatment effects [3,22,24]. The therapeutic effects at the cellular level were then amplified and clinically maximized by performing confluent treatment of large areas of diseased, dysfunctional retina [3]. For example, rather than focal treatment of retinal microaneurysms in diabetic macular edema, treatment of the entire area of macular thickening and eventually the entire posterior retina (“panmacular” treatment) was employed, recognizing the need to more fully address the more widespread nature of the disease process [22,23,24].

Without LIRD and its inflammatory response, SDM does not acutely resolve drusen [3,6]. Clinically and in laboratory studies, 810 nm MPL such as SDM reduces, rather than incites, inflammation [3]. By exclusion, in the absence of LIRD, the therapeutic response to retinal laser could only arise from retina affected but not killed by treatment [3,6,22,23,24]. Further, rather than compromising retinal function via LIRD, SDM preserved and improved visual and retinal function [3,13]. This represents a physiologic “reset” of retinal function, reflecting low-dose adaptive thermal hormesis triggered by laser-induced RPE heat-shock protein (HSP) activation, normalizing proteostasis and inhibiting apoptosis to improve all aspects of RPE, and hence retinal function [3,13]. In conception and treatment application, the absence of the anatomic alterations resulting from LIRD makes SDM more akin to a medical treatment rather than a surgical procedure, and thus a notable departure from the laser approaches previously applied to dry AMD [3] (Figure 3).

### 3.2. SDM for Dry AMD

In 2016, it was reported that within a month of panmacular SDM treatment in eyes with dry AMD and inherited retinal degenerations, electrophysiology by pattern electroretinography (PERG) was improved in 138/156 eyes (88%) tested with dry AMD (*p* = 0.00001) and in 10/10 eyes with IRDs, including Stargardt’s disease and retinitis pigmentosa (*p* = 0.002) [13]. Microperimetry (*p* = 0.04) and mesopic contrast VA were also improved (*p* = 0.006) [13,28,29,30]. Linear regression analysis found that the eyes with the greatest degree of pre-treatment dysfunction improved the most following treatment. These findings were consistent with the “reset” theory of laser action that hypothesized treatment-induced normalization of retinal function [3,6,13]. The study noted that as improved function was a surrogate for slowed disease progression, the results suggested that periodic re-treatment as maintenance therapy might reduce the risks of visual loss in AMD [3,6,13,28,29,30].

In a subsequent retrospective case–control study, between 2014 and 2018, regular periodic SDM as maintenance therapy (“vision protection therapy”, VPT), informed by the significant improvements in retinal and visual function reported following SDM for dry AMD as surrogates for long-term risk reduction, was offered to 95% of all patients in a vitreoretinal subspecialty practice with dry AMD of Age-Related Eye Disease Study (AREDS) category 2 or greater disease in at least one eye [9,31]. Reported in 2018, treatment was elected by 97%, and of these, follow-up was available for 98% of treated eyes, or 547 eyes of 348 patients [31]. SDM was performed on average every 8 months. Post-treatment follow-up averaged 22 months. In that interval, patients were followed with clinical examination, fundus photography, including fundus autofluorescence fundus photography, and optical coherence tomography (OCT) every 3–4 months. The primary study endpoint was neovascular conversion indicated by the development of macular exudation on OCT and confirmed by fundus fluorescein and/or indocyanine green angiography. Of note is that this cohort had exceptionally high risk factors for conversion to NAMD, including advanced age (median 84 years), AREDS category (78% categories 3 and 4), fellow eye NAMD (128 eyes, 23%), and reticular pseudodrusen (214 eyes, 39%) [3,6,9,30,31]. In these eyes managed with VPT, neovascular conversion was observed in 9/547 eyes (1.6% total, or 0.87%/year) with the highest-risk eyes benefiting the most from treatment. Adjusting the conversion risk to reflect the large difference in age between the study group and AREDS patients (AREDS avg. age 69 years, 15 years younger than the study population) yielded an age-adjusted reduction in the rate of neovascular conversion of 93–98% per year, compared to the AREDS neovascular conversion rate reduction of approximately 4% per year [31].

### 3.3. SDM for Geographic Atrophy (GA)

In this same cohort of high-risk dry AMD, 67 eyes of 49 patients with GA were identified as having photographic and OCT documentation both before and after initiation of VPT [32]. These high-risk eyes, avg. age 86 years, with RPD in 52/67 (78%), were observed a mean of 2.5 years prior to the initiation of VPT and followed up a mean of 2.2 years after initiation of VPT.

Prior to initiation of VPT, the average rate of linear radial GA progression was 137 μm/year, comparable to natural history studies of GA [32]. After initiation of VPT (SDM performed on average every 112 days), overall progression slowed to an average of 73 μm/year, representing a 47% reduction in the annual rate of progression (*p* < 0.0001). GA lesions < 1000 μm in diameter, although too few for statistical significance, slowed nearly twice as much. Despite high risk factors, no eye developed NAMD after initiation of VPT [32] (Figure 3). Thus, VPT for GA in this study was notably both safer and more effective than either retinal-damaging laser modes or currently available targeted drug therapies such as complement fixation inhibitor intravitreal injections, which achieve less slowing of GA progression while substantially increasing the risk of neovascular conversion [3,32,33,34,35].

### 3.4. Real-World Data Studies of SDM VPT for Dry AMD

Vestrum Health, Inc. (Naperville, IL, USA), aggregates unidentified patient data harvested directly from electronic medical records (EMRs) from over 300 retina subspecialty practices in the United States. Vestrum data has been used for a number of RWD studies, such as examinations of clinical use of anti-VEGF therapy for AMD and diabetic retinopathy [35,36]. The large retina-centric Vestrum database of unidentified patient EMR data was thus selected to attempt a comparison of the results of SDM VPT for dry AMD to standard care with AREDS supplements alone for the prevention of neovascular conversion. Propensity scoring (PS) is a statistical method that achieves comparisons between groups by first matching and then randomizing comparisons of matched individuals from an existing population, such as a large database (RWD), rather than from an acquired population (RCT) [37,38,39,40,41,42].

Two studies on the effect of SDM VPT on the rate of neovascular conversion in dry AMD were reported using PS analyses of real-world data from the Vestrum database. These studies shared several similarities [43,44]. In both studies, all data was obtained from aggregated unidentified patient EMR data harvested from the Vestrum database and propensity scored by Vestrum for all patients in the database coded for dry AMD. Only patients with bilateral dry AMD were included. Eyes with other significant or confounding macular or ocular disease, including prior anti-VEGF injections, were excluded. Neovascular risk factors such as patient age, sex, use of AREDS supplements, smoking history, and systemic hypertension were recorded. To create a binary comparison, Vestrum-recorded data on all dry AMD patients in the practice routinely employing VPT was compared to all dry AMD patients in the rest of the Vestrum panel after excluding all eyes with prior or current macular laser “standard care alone” or “SCA” group). In both studies, two-factor confirmation of neovascular conversion was employed, consisting of a change of diagnostic coding from dry to wet, or NAMD, as well as coincident initiation of anti-VEGF therapy [43,44]. In both studies, the VPT groups were treated with panmacular SDM performed every 3–4 months employing identical laser parameters and numbers of laser spots applied confluently to the retina between the major vascular arcades (“panmacular”) in every eye of every treated patient. SDM is thus a highly uniform dose-like treatment minimizing the influence of surgeon skill and experience [43,44].

Study differences: The principal difference between the two PS RWD studies comparing VPT to SCA for the prevention of NAMD was the longitudinal time frame of each study and ICD (International Coding of Disease) coding conventions for AMD utilized at the time [43,44]. The first study (PS1), between 4 January 2016 and 30 September 2020, included the use of ICD-9 codes which distinguished only between dry and wet AMD [43]. The second study (PS2), examining data collected between January 2017 and July 2023, included only ICD 10 codes that stratified dry AMD into four different levels of increasing disease severity (and thus neovascular conversion risk) [44].

SDM VPT RWD study results: Large numbers of eyes in the Vestrum database were coded for dry AMD (392,000 eyes in PS1 and 500,000 in PS2). These populations were filtered for inclusion and exclusion criteria, then propensity scored and randomly matched by Vestrum in a 10/1 ratio (SCA/VPT eyes) for statistical comparison, resulting in 8300 SCA and 830 VPT eyes for comparison in PS1 and 7370 SCA and 737 VPT eyes in PS2 [43,44]. Quality checks including computational modeling showed high levels of matching concordance and successful tests of statistical validity in both studies [43,44]. A total of six PS analyses between the two studies using different data permutations were performed, finding the same results, a strong indicator of study reliability. Quintile analysis of each of the six PS analyses, able to detect 99% of unrecognized biases, revealed no evidence of undetected bias in any of the analyses [37,38,39,40,41,43,44] (Figure 4).

Both PS RWD studies confirmed the findings of the retrospective consecutive case series, demonstrating that VPT markedly reduced the risk of neovascular conversion in dry AMD compared to SCA (HR 13 and HR 6, respectively) [43,44] (Figure 5). Like the earlier retrospective study, ICD 10 code analysis in PS2 found that the highest-risk eyes benefited most from VPT [43,44]. Of note was that the advantage of VPT in both studies, but particularly PS2, was muted by the failure of SCA to identify a high percentage of neovascular conversions [43,44]. This was identified statistically and attributed to the low encounter (clinical examination) frequency in the SCA group, generally examined annually or less, compared to the VPT group, which was examined on average every 108 days [43,44]. In PS1, statistically balancing the encounter frequency increased the advantage of VPT over standard care from HR 8 to HR 13, indicating that VPT-treated eyes were 13× less likely on a daily basis to experience neovascular conversion than eyes managed with AREDS supplements alone [43]. In PS2, the examination frequency in the SCA group was too low to fully statistically balance with the VPT group [44]. However, by extrapolating the 1% increase in observed conversions for every 1% increase in SCA encounters, the projected advantage of VPT over SCA in PS2 doubled to approximately HR12, similar to PS1 [43,44] (Figure 4 and Figure 5).

In these PS RWD studies, VPT was continued in the VPT group eyes if neovascular conversion occurred, based on prior studies showing that SDM could improve the performance of anti-VEGF therapy by preventing drug tolerance [43,44,45]. Analysis showed that eyes continuing VPT after neovascular conversion required 3.9–4.5× fewer injections than eyes in the SCA group managed with injections alone, while also maintaining better vision than converted eyes treated with drug monotherapy [3,6,44,45].

### 3.5. Photobiomodulation

While not properly laser treatment, photobiomodulation (PBM) has recently received attention for the treatment of dry AMD [46]. Popular for many years, especially in Eastern Europe, the use of visible light exposure, commonly referred to as “red light therapy”, has recently been approved by the United States Food and Drug Administration (FDA) for the treatment of dry AMD [47]. The simplest distinction between retinal laser treatment and PBM is that retinal laser is thermal, while PBM is not. Unlike retinal laser treatment whose therapeutic effects are mediated by laser-induced retinal hyperthermia causing low-dose adaptive hormesis in the retina initiated by RPE HSP activation (the catabolic phase of the hormetic response) resulting in normalized proteostasis and thus normalized cell function (the anabolic phase of the hormetic response), reduced inflammation, and therapeutic immunoactivation, PBM operates via an entirely different mechanism and thus has a different and generally more narrow range of effects [3,6,13,24,45,46,47,48,49]. In PBM, photons from visible light, generally produced by incandescent bulbs or light-emitting diode sources, are absorbed by electrons in the metal cations of the respiratory chain molecules, energetically promoting them temporarily into a higher energy valence. Just as quickly, the electron orbit degrades to return to its native state, releasing the absorbed energy into the system. This is termed the “photoelectric effect” [3,48]. The principal result is to temporarily improve ATP production and mitochondrial function. The effects of non-thermal PBM are highly wavelength-dependent and order-of-presentation-specific, while the thermal effects of retinal laser are not [3,46,47,48]. Because of the different mechanisms of action, the effects of retinal laser tend to be more therapeutically diverse, longer-acting, and more robust [3,49].

In the Lightsite III trial examining the use of the Valeda PBM device (Valeda Light Delivery System (LumiThera, Inc., Poulsbo, WA, USA), Valeda PBM-treated eyes were compared with “sham” treatment (lower-intensity illumination) over 13 months [47]. A total of 100 patients (148 eyes) were eligible for randomization. A total of 91 treated and 54 sham eyes entered the study; however, the planned drop-out rate of 10% was exceeded, as 31% of randomized patients failed to complete the study that required an onerous nine treatment sessions over 3–5 weeks every 4 months, resulting in just 79 treated and 40 control eyes completing the study. FDA approval appeared to rely on a modest 2.4-letter average visual acuity improvement in treated patients (*p* = 0.02) over controls at the end of 13 months. Secondary outcomes including low-light best-corrected VA, Radner reading chart performance, and Visual Quality of Life (VFQ)-25 scores were unchanged in both treated and control eyes. The development of new GA appeared to be reduced in treated eyes (*p* = 0.0024) compared to controls. However, while the authors describe a “favorable safety profile”, three times as many treated eyes developed neovascular AMD as controls (5.4 vs. 1.8%), a serious adverse treatment effect suggesting the possibility of phototoxicity [47]. Thus, despite regulatory approval, questions remain and concerns have been expressed regarding the Lightsite III trial and the efficacy and safety of PBM for dry AMD [50].

The paucity of studies of retinal laser for prophylaxis of AMD since the “laser for drusen” studies of the late 1990s is notable. This reflects two major influences. First, it was found that conventional CW and subsequent short-pulse laser modes not only failed to slow disease progression to prevent vision loss, but often worsened and accelerated progression and vision loss, especially in higher-risk eyes most in need of aid [3,5,6,12,14,15,16,17,18,19]. The constant in these attempts was the determination to eliminate drusen and causation by design of LIRD to induce macular inflammation required to achieve this end [3,6,8,12,14,15,16,17,18,19]. Second, the rapid rise of drug therapy in ophthalmology began on the temporal heels of these laser failures, further souring interest in retinal laser treatment in general and shifting hopes toward future safer, more effective drug therapies [33,34,35,36,51,52]. However, the results of alternative treatments such as targeted drug therapy and photobiomodulation have thus far demonstrated only modest therapeutic benefits while being burdensome, invasive, and expensive in the case of drug therapy, with both increasing the risk of neovascular conversion, arguably the most severe adverse outcome from AMD [32,33,34,47,50,53]. With time, these may improve, and new treatments will arrive. However, current evidence indicates that macular laser treatment is the most promising treatment of any kind for preventing visual loss from dry AMD. This reflects a strategy of using retinal lasers to transform and functionally normalize rather than destroy diseased retinas, which has shown robust and consistent therapeutic responses in all chronic progressive retinopathies studied thus far, including AMD [3]. The simplicity, uniformity, safety, durability, and low cost of laser therapy for dry AMD, such as SDM VPT, recommend further study. If currently available data can be confirmed, macular laser prophylaxis of dry AMD would become the most important clinical measure to prevent vision loss from AMD, the most common cause of irreversible vision loss worldwide [1,3,6,31,32,43,44,45]. Unfortunately, largely due to lack of financial sponsorship, no such study is on the horizon [53,54,55,56]. Retina-damaging laser treatments for dry AMD should clearly be avoided [3,5,6]. However, available data, the safety and uniformity of treatment, the significant need, and the absence of other comparably safe and effective interventions suggest that it may not be unreasonable to consider SDM VPT for patients with AMD who are sufficiently anxious about or at high risk for age-related vision loss who may not be willing, or indeed able, to wait for years for confirmatory RCTs, which may or may not come at all [57].

## 4. Conclusions

Current evidence indicates that retinal laser treatment is the safest and most effective treatment of any kind to prevent progression and vision loss in dry AMD. The safety and effectiveness of macular laser for dry AMD appear to hinge on treatment wholly sublethal to the RPE applied widely over the macula to optimize the clinical treatment benefits conducted on a regular basis to maintain the treatment benefits over time [58]. This treatment approach is currently exemplified by panmacular low-intensity/high-density subthreshold diode micropulse laser (SDM) performed as vision protection therapy. Further studies are needed, particularly RCTs, to confirm current evidence and increase our knowledge to optimize treatment parameters, schedules, indications, and expectations. In view of the vast numbers of patients at risk for vision loss from AMD, treatment automation is needed to improve accessibility and efficiency, maximize patient acceptance, and minimize risk. The potential value to patients and public health from prophylactic macular laser for AMD suggests that these needs should be given a higher priority than currently afforded.

## Figures and Tables

**Figure 1 jcm-14-04157-f001:**
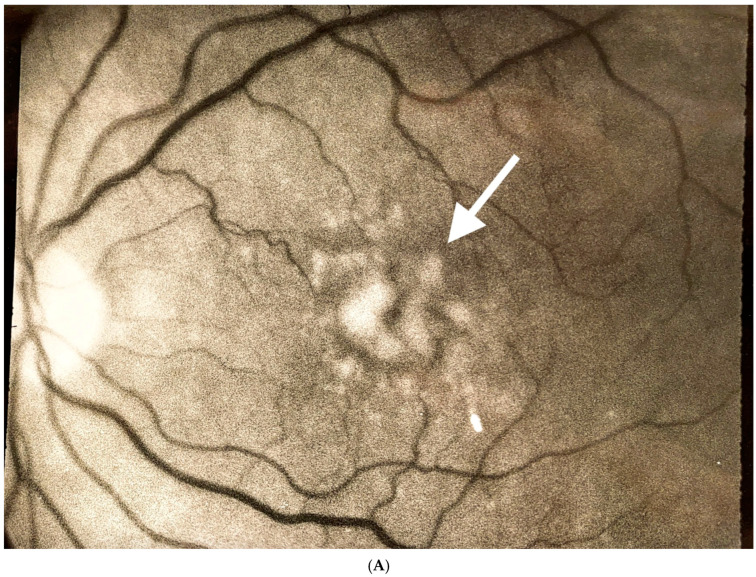
Eye with intermediate dry age-related macular degeneration. (**A**) Fundus photograph before “subthreshold” laser treatment to reduce drusen. Arrow identifies large “soft-type” drusen. (**B**) Intravenous fundus fluorescein angiograph after treatment. Note laser-induced retinal damage (arrow). (**C**) Fundus photograph after laser treatment. Note the resolution of macular drusen (arrow). Courtesy of Luttrull, J.K. *Modern Retinal Laser Therapy. Principles and Application.* Kugler Publications, Amsterdam, The Netherlands August 2023 ISBN: 978-90-6299298-0 [3].

**Figure 2 jcm-14-04157-f002:**
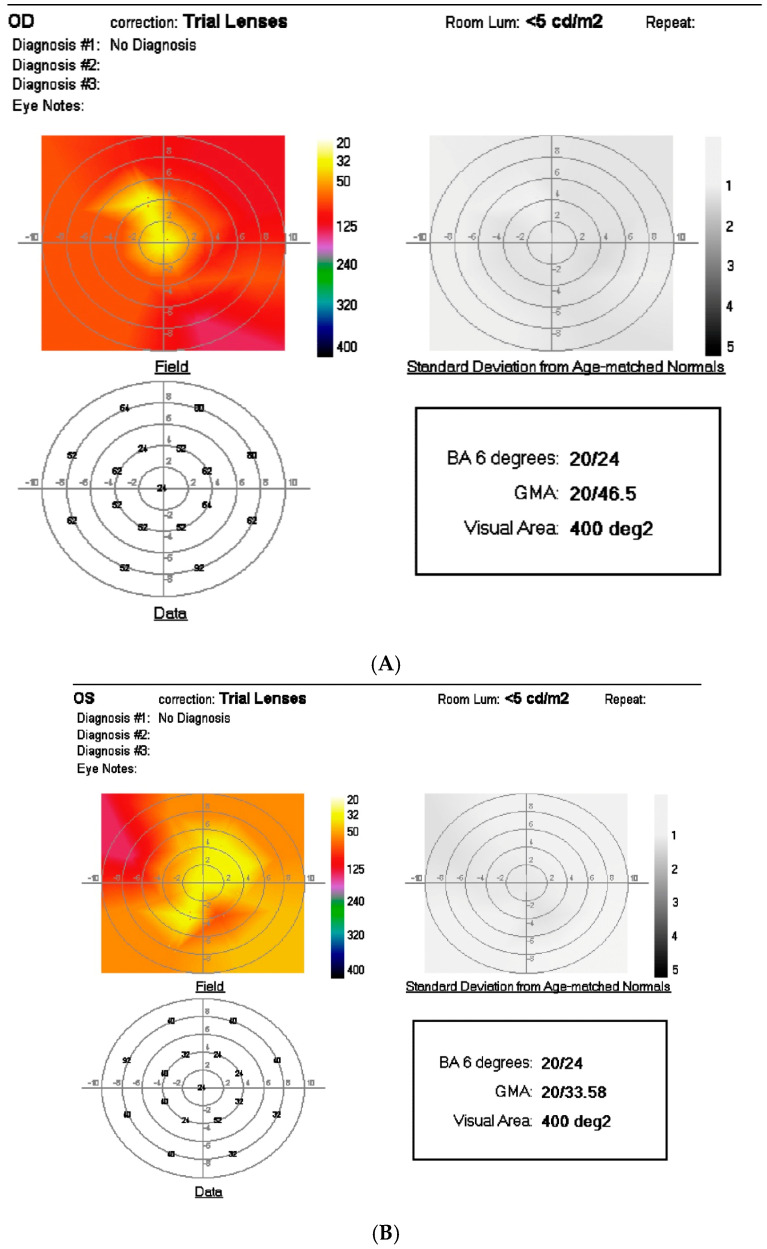
Mesopic 10-degree visual field before (**A**) and after (**B**) SDM for dry AMD. BA 6 = best mesopic visual acuity within 6 degrees of fixation. GMA = average mesopic visual acuity within central 10 degrees. Visual area = degrees of measurable mesopic visual function within 20 degrees of fixation. Note improvement in mesopic visual function. Scatter graph (**C**) showing pattern electroretinogram responses before and after SDM for dry AMD. Points above the diagonal line denote improvement; points below indicate worsening post-treatment. Note improvement in the low-contrast latency signal measure after treatment (*p* = 0.001). Dark adaptometry before (**D**,**E**) and after SDM for dry AMD in the left eye only. The steeper slope of dark adaption recovery over time denotes more normal macular function. Note the increase in the slope of the blue stimulus response in the left eye only indicating improved perifoveal rod function in the left eye following SDM treatment. (**A**,**B**,**D**,**E**): from Luttrull, J.K. *Modern Retinal Laser Therapy. Principles and Application*. Kugler Publications, Amsterdam, The Netherlands August 2023 ISBN: 978-90-6299298-0. (**C**): from Luttrull J.K.; Margolis, B.W.L. Functionally guided retinal protective therapy as prophylaxis for age-related and inherited retinal degenerations. A pilot study. *Investig. Ophthalmol. Vis. Sci*. **2016**, *5*, 265–275. https://doi.org/10.1167/iovs.15-18163 [3,13].

**Figure 3 jcm-14-04157-f003:**
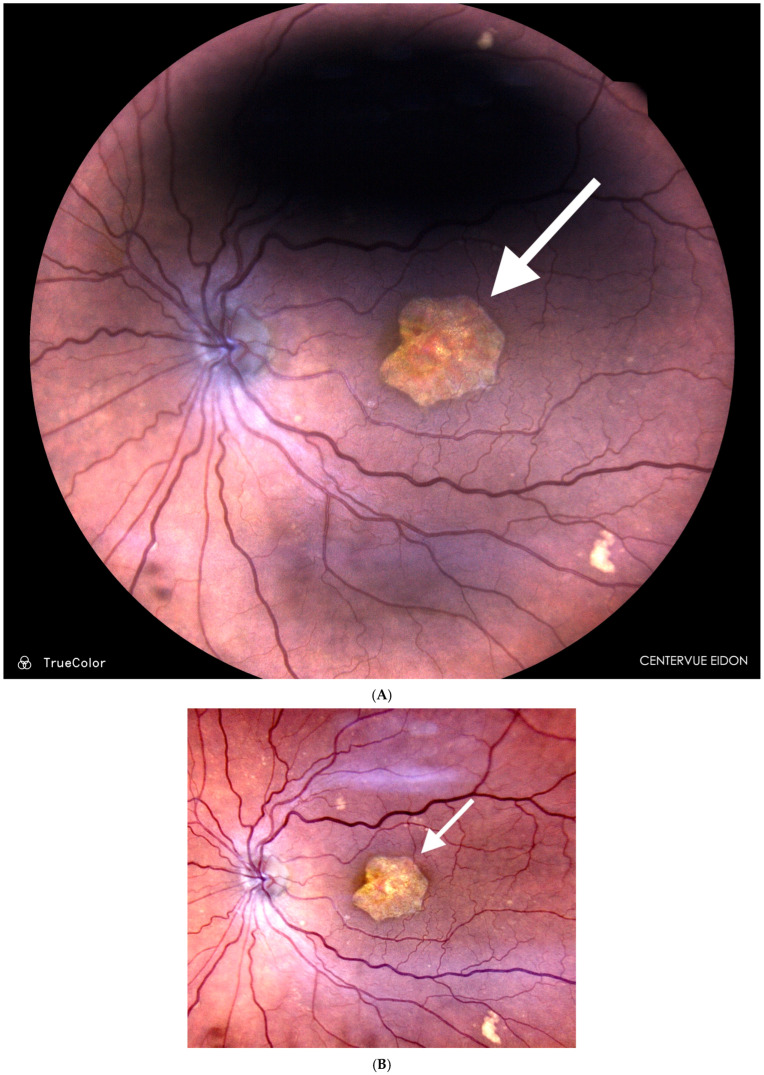
Slowing of geographic atrophy (GA) and improved visual acuity (VA) following SDM VPT in dry AMD. Fundus photographs of an 85-year-old pseudophakic patient with open-angle glaucoma and central geographic atrophy (arrow) with VA OS 20/200 since 2016 when SDM vision protection therapy was started to slow progression. (**A**) January 2021, VA still 20/200. VA then gradually improved until the most recent visit in June 2024, VA 20/40 + 2 (**B**). GA lesion diameter in January 2021 was 2578 μm and in June 2024 it was 2896 μm with an annual radial linear progression rate of 45 μm/year (natural history expected 137 μm/year), or a 67% lower rate of GA progression compared to the average expected rate of progression. In addition to slowing GA progression, this case illustrates the common observation that, by improving macular function, SDM often improves visual function even in the absence of notable anatomic change in all applications.

**Figure 4 jcm-14-04157-f004:**
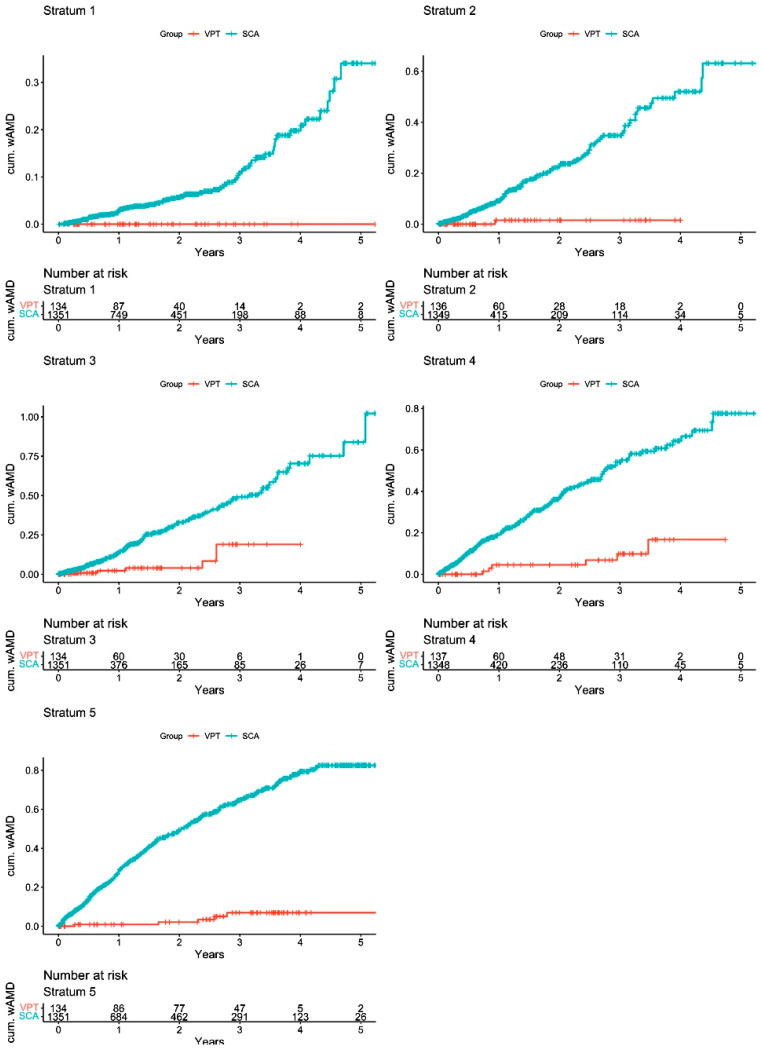
Quintile analysis graphs of cumulative probability of neovascular AMD conversion by propensity score strata 1–5 from Luttrull, J.K.; Gray, G.; Bhavan, S.V. Vision protection therapy for prevention of neovascular age-related macular degeneration. *Sci. Rep.* **2023**, *13*, 16710. https://doi.org/10.1038/s41598-023-43605-w [44]. The survival analysis was stratified by propensity score quintiles by dividing patients into five nearly equal-sized groups using the quintiles of the propensity scores, from the lowest (first quintile) to highest (fifth quintile) degree of risk-factor matching. The plots show the survival curves by propensity score stratum. Note that for every risk-factor-matched quintile, VPT significantly reduced the rate of neovascular conversion compared to standard care alone. Agreement between all 5 quintiles eliminates 99% of any undetected biases in study construction. VPT = vision protection therapy; SCA = standard care alone; AMD = age-related macular degeneration; Cum = cumulative.

**Figure 5 jcm-14-04157-f005:**
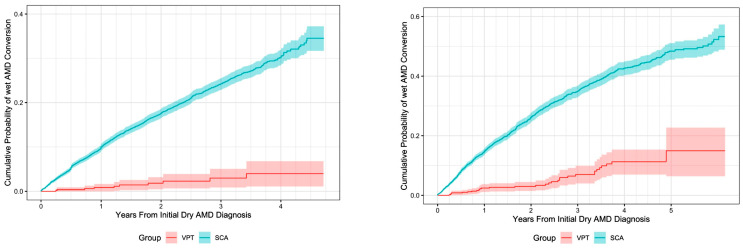
Overall Kaplan–Meier cumulative wet AMD conversion probability by group, ignoring covariates from propensity-score-analyzed Vestrum real-world data studies, comparing vision protection therapy to standard antioxidant vitamin therapy alone, with encounter matching. Shaded areas indicate 95% confidence intervals. Both graphs (left and right) show the marked and ever-increasing advantage of SDM VPT compared to standard care alone over time in reducing the rate of neovascular conversion in dry AMD. AMD = age-related macular degeneration; SCA = standard care alone; VPT = vision protection therapy. The graph on the left depicts an advantage of VPT over SCA (AREDS vitamins alone) of a hazard ratio (HR) of 13. The graph on the left depicts an advantage of VPT over SCA (AREDS vitamins alone) of a hazard ratio (HR) of 6. However, the HR of 6 from the subsequent study was found to underestimate the advantage of VPT by approximately half due to the high percentage of missed neovascular conversions in the SCA group resulting from infrequent clinical examinations. Left, from Luttrull, J.K.; Gray, G. Real World Data Comparison of Standard Care vs. SDM Laser Vision Protection Therapy for Prevention of Neovascular AMD. *Clin. Ophthalmol.* **2022**, *16*, 1555–1568. Right, from Luttrull, J.K.; Gray, G.; Bhavan, S.V. Vision protection therapy for prevention of neovascular age-related macular degeneration. *Sci. Rep.* **2023**, *13*, 16710. https://doi.org/10.1038/s41598-023-43605-w) [43,44].

## Data Availability

All data are publicly available in peer-reviewed publications.

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
