# Peer review of "Laser Prophylaxis for Dry Age-Related Macular Degeneration: Current Evidence"

_jcm, 2025, doi:10.3390/jcm14124157_

Round 1

Reviewer 1 Report

Comments and Suggestions for Authors The paper covers the topic that has been very important for years - laser treatment for non-exudative AMD. The abstract is clear and states the purpose of the research. The manuscript is correct in terms of the structure and the merits. This is a narrative review, discussion is well argued and clear. The authors have presented in detail various laser types and their treatment effects on dry AMD and geographic atrophy. Sections are coherent and guide readers effectively through the subject. The review covers a wide range of relevant studies and presents them in a cohesive manner. The references in this paper are reasonable and closely integrated with the content of the whole paper. However, a few elements could be improved:
  1. Methodology: a brief description of the search strategy for literature selection with inclusion and exclusion criteria would help improve transparency.
  2. Figures 1.A-C are of low quality with motion artefacts. The authors should provide clearer eye fundus photos to better illustrate the pathology.
  3. A conclusion section summarizing the discussion should be added to emphasize the future directions regarding laser therapy for AMD.

Author Response

The paper covers the topic that has been very important for years - laser treatment for non-exudative AMD. The abstract is clear and states the purpose of the research. The manuscript is correct in terms of the structure and the merits. This is a narrative review, discussion is well argued and clear. The authors have presented in detail various laser types and their treatment effects on dry AMD and geographic atrophy. Sections are coherent and guide readers effectively through the subject. The review covers a wide range of relevant studies and presents them in a cohesive manner. The references in this paper are reasonable and closely integrated with the content of the whole paper. However, a few elements could be improved:

  1. Methodology: a brief description of the search strategy for literature selection with inclusion and exclusion criteria would help improve transparency.

To address the reviewer’s concern we have expanded the Methods lines 101-112. “A literature review based on studies of macular laser treatment for dry AMD listed in the PubMed database. http://pubmed.ncbi.nlm.nih.gov/  . Search terms included macula, laser, age-related macular degeneration, subthreshold, prevention, and prophylaxis.  Primary studies reporting clinical outcomes of macular laser for dry AMD were included, except in the case of the prior “laser for drusen” studies where the large number of studies recommended use of meta-analyses, such as the Cochrane Database. 5 Secondary and sub-analyses, and reviews were not included. 173 items were returned in PubMed based on the key word search. Despite the query specifying age-related macular degeneration, the majority of search returns described laser treatment of diabetic macular edema. Excluding the “laser for drusen” studies included in the 2015 Cochrane Database meta-analysis, 9 subsequent primary studies reporting the clinical outcomes of prophylactic macular laser for dry AMD were identified for inclusion in this report.5  “

  1. Figures 1.A-C are of low quality with motion artefacts. The authors should provide clearer eye fundus photos to better illustrate the pathology.

These are archival photos from the pre-digital age when “laser for drusen” was being done. We would also like better resolution, but it is what it is. Despite the imperfection one can easily druse before treatment, resolution of drusen after treatment, and laser damage that cause the drusen to resolve. That is the only point of the figure. Arrows have been added to the figures to help orient readers to the points of interest.

  1. A conclusion section summarizing the discussion should be added to emphasize the future directions regarding laser therapy for AMD.

To address the reviewer’s concern we have added the follow paragraph, entitled “Conclusion”, lines 450-463:

Conclusion

Current evidence indicates that retinal laser treatment is the safest and most effective treatment of any kind to prevent progression and vision loss dry AMD. The safety and effectiveness of macular laser for dry AMD appear to hinge on treatment wholly sublethal to the RPE applied widely over the macular to optimize the clinical treatment benefits done on a regular basis to maintain the treatment benefits over time. This treatment approach is currently exemplified by panmacular low-intensity / high-density subthreshold diode micropulse laser (SDM) performed as vision protection therapy. Further studies are needed, particularly RCTs, to confirm current evidence and increase our knowledge to optimize treatment parameters, schedules, indications, and expectations. In view of the vast numbers of patients as risk for vision loss from AMD, treatment automation is needed to improve accessibility, efficiency, maximize patient acceptance, and minimize risk. The potential value to patients and public health from prophylactic macular laser for AMD suggest that these needs should be given a higher priority than currently afforded.

Reviewer 2 Report

Comments and Suggestions for Authors

The manuscript is interesting. This summary will be helpful to understand the laser prophylaxis for AMD patients. Several comments will make the manuscript have more impact and clear conclusions.

The method section should be well improved. It is written that PubMed was used. However, how it was used should be well-described. For example, some keywords (dry AMD, laser, etc) have been searched and we collected XXX papers and several papers was excluded because of YYY reasons. Therefore, finally XXX-ZZZ papers have been reviewed and summarized in this manuscript. Additionally, it is good to draw a process how it was collected.

Each image on Figure 1 should have some indications for the damaged areas or emphasized locations. Without the indications, the future readers may be difficult to understand what to see or where to check.

If Figure 2 is used and presented in the manuscript, each section and image should be kindly discussed. Without this, it is not so meaningful as a figure. It should be removed in that case. Additionally, each graph's x-axis and y-axis and some other presented information explanations are needed.

Figure 3 needs the same request (like Figure 1). 

Figure 4 and 5 need the same request (like Figure  2). 

Basically, results and discussions can be combined but in this manuscript, the future direction and limitations of the study are missing with the conclusion part. It is recommended to include the conclusion part separately. 

In the introduction, it might be important to cover the pathology of AMD and current treatment shortly to make a good flow of the review work for laser prophylaxis for AMD cases. 

Author Response

Reviewer 1:

The paper covers the topic that has been very important for years - laser treatment for non-exudative AMD. The abstract is clear and states the purpose of the research. The manuscript is correct in terms of the structure and the merits. This is a narrative review, discussion is well argued and clear. The authors have presented in detail various laser types and their treatment effects on dry AMD and geographic atrophy. Sections are coherent and guide readers effectively through the subject. The review covers a wide range of relevant studies and presents them in a cohesive manner. The references in this paper are reasonable and closely integrated with the content of the whole paper. However, a few elements could be improved:

  1. Methodology: a brief description of the search strategy for literature selection with inclusion and exclusion criteria would help improve transparency.

To address the reviewer’s concern we have expanded the Methods lines 101-112. “A literature review based on studies of macular laser treatment for dry AMD listed in the PubMed database. http://pubmed.ncbi.nlm.nih.gov/  . Search terms included macula, laser, age-related macular degeneration, subthreshold, prevention, and prophylaxis.  Primary studies reporting clinical outcomes of macular laser for dry AMD were included, except in the case of the prior “laser for drusen” studies where the large number of studies recommended use of meta-analyses, such as the Cochrane Database. 5 Secondary and sub-analyses, and reviews were not included. 173 items were returned in PubMed based on the key word search. Despite the query specifying age-related macular degeneration, the majority of search returns described laser treatment of diabetic macular edema. Excluding the “laser for drusen” studies included in the 2015 Cochrane Database meta-analysis, 9 subsequent primary studies reporting the clinical outcomes of prophylactic macular laser for dry AMD were identified for inclusion in this report.5  “

  1. Figures 1.A-C are of low quality with motion artefacts. The authors should provide clearer eye fundus photos to better illustrate the pathology.

These are archival photos from the pre-digital age when “laser for drusen” was being done. We would also like better resolution, but it is what it is. Despite the imperfection one can easily druse before treatment, resolution of drusen after treatment, and laser damage that cause the drusen to resolve. That is the only point of the figure. Arrows have been added to the figures to help orient readers to the points of interest.

  1. A conclusion section summarizing the discussion should be added to emphasize the future directions regarding laser therapy for AMD.

To address the reviewer’s concern we have added the follow paragraph, entitled “Conclusion”, lines 450-463:

Conclusion

Current evidence indicates that retinal laser treatment is the safest and most effective treatment of any kind to prevent progression and vision loss dry AMD. The safety and effectiveness of macular laser for dry AMD appear to hinge on treatment wholly sublethal to the RPE applied widely over the macular to optimize the clinical treatment benefits done on a regular basis to maintain the treatment benefits over time. This treatment approach is currently exemplified by panmacular low-intensity / high-density subthreshold diode micropulse laser (SDM) performed as vision protection therapy. Further studies are needed, particularly RCTs, to confirm current evidence and increase our knowledge to optimize treatment parameters, schedules, indications, and expectations. In view of the vast numbers of patients as risk for vision loss from AMD, treatment automation is needed to improve accessibility, efficiency, maximize patient acceptance, and minimize risk. The potential value to patients and public health from prophylactic macular laser for AMD suggest that these needs should be given a higher priority than currently afforded.

Reviewer 2:

The manuscript is interesting. This summary will be helpful to understand the laser prophylaxis for AMD patients. Several comments will make the manuscript have more impact and clear conclusions.

The method section should be well improved. It is written that PubMed was used. However, how it was used should be well-described. For example, some keywords (dry AMD, laser, etc) have been searched and we collected XXX papers and several papers was excluded because of YYY reasons. Therefore, finally XXX-ZZZ papers have been reviewed and summarized in this manuscript. Additionally, it is good to draw a process how it was collected.

We thank the Reviewer for the suggestion.  Please see the response to Reviewer 1 query number 1 above.

Each image on Figure 1 should have some indications for the damaged areas or emphasized locations. Without the indications, the future readers may be difficult to understand what to see or where to check.

We thank the Reviewer for the suggestion. Arrows have been added to the figures and defined in the Legend to aid readers.

If Figure 2 is used and presented in the manuscript, each section and image should be kindly discussed. Without this, it is not so meaningful as a figure. It should be removed in that case. Additionally, each graph's x-axis and y-axis and some other presented information explanations are needed.

Additional explanatory info has been added to the figure legend.

Figure 3 needs the same request (like Figure 1). 

Arrows have been added to identify GA

Figure 4 and 5 need the same request (like Figure  2). 

Figure 4: We believe the Legend for Figure 4 is clear and complete as stated.

Figure 5: The Legend has been amplified with the following: “The graph on the left depicts an advantage of VPT over SCA (AREDS vitamins alone) of a hazard ratio (HR) of 13. The graph on the left depicts an advantage of VPT over SCA (AREDS vitamins alone) of a hazard ratio (HR) of 6. However, the HR of 6 in the subsequent study was found to underestimate the advantage of VPT by approximately half due to the high percentage of missed neovascular conversions in the SCA group resulting from infrequent clinical examinations.”

Basically, results and discussions can be combined but in this manuscript, the future direction and limitations of the study are missing with the conclusion part. It is recommended to include the conclusion part separately.

Please see responses to Reviewer 1 above 

In the introduction, it might be important to cover the pathology of AMD and current treatment shortly to make a good flow of the review work for laser prophylaxis for AMD cases.

To address the Reviewer’s request we have added the following paragraph to the Introduction, lines 79-88: “As a chronic progressive retinopathy (CPR), AMD is by definition a neurodegeneration and thus part of the predominant class of disease affecting the aging population worldwide. 1,3,6 The most important predispositions to AMD, beside age, include race, nutrition, life-style habits such as smoking, and genetic profiles. Like all diseases of aging, the course of AMD is characterized by cumulative cellular dysfunction leading to anatomic disruption and degeneration fomenting chronic inflammation. 3,6 In the advanced endstages of AMD vision loss may ensue due to development of macular geographic chorioretinal atrophy (GA) or NAMD. Historically, the principal therapy for dry AMD has been use of anti-oxidant vitamins and lifestyle adjustment such as improved diet and smoking avoidance whose success can best be described as modest.1-10

Reviewer 3 Report

Comments and Suggestions for Authors

Dear Author

Thank you for the very well written review about the current evidence for Laser Prophylaxis for dry AMD.  Apart from e few minor questions/ typographical errors (see below) I do have one major question: Is there a specific reason why photobiomodulation (Boyer D, Hu A, Warrow D, Xavier S, Gonzalez V, Lad E, Rosen RB, Do D, Schneiderman T, Ho A, Munk MR, Jaffe G, Tedford SE, Croissant CL, Walker M, Rückert R, Tedford CE. LIGHTSITE III: 13-Month Efficacy and Safety Evaluation of Multiwavelength Photobiomodulation in Nonexudative (Dry) Age-Related Macular Degeneration Using the Lumithera Valeda Light Delivery System. Retina. 2024 Mar 1;44(3):487-497. doi: 10.1097/IAE.0000000000003980. PMID: 37972955; PMCID: PMC10885856.) is not included in the review? This is, according to my knowledge another approach to -broadly speaking- laser-light prophylaxis for dry AMD. 

Introduction

General: I have not seen you introduce the abbreviation SDM. 

Line 45: there is no therapy without possible side effects, thus “can be employed as a medical therapy without adverse effects” seems somewhat of an exaggeration.

Results/Discussion

Line 63: dating back several millennia – including the treatment of the eye? Is that really true? Several millennia would imply some 3000+years back. Please reword if the millennia only concern other applications of lasers.

Line 184. This represents (not represent)

Author Response

Dear Author

Thank you for the very well written review about the current evidence for Laser Prophylaxis for dry AMD.  Apart from e few minor questions/ typographical errors (see below) I do have one major question: Is there a specific reason why photobiomodulation (Boyer D, Hu A, Warrow D, Xavier S, Gonzalez V, Lad E, Rosen RB, Do D, Schneiderman T, Ho A, Munk MR, Jaffe G, Tedford SE, Croissant CL, Walker M, Rückert R, Tedford CE. LIGHTSITE III: 13-Month Efficacy and Safety Evaluation of Multiwavelength Photobiomodulation in Nonexudative (Dry) Age-Related Macular Degeneration Using the Lumithera Valeda Light Delivery System. Retina. 2024 Mar 1;44(3):487-497. doi: 10.1097/IAE.0000000000003980. PMID: 37972955; PMCID: PMC10885856.) is not included in the review? This is, according to my knowledge another approach to -broadly speaking- laser-light prophylaxis for dry AMD. 

Because PBM is not retinal laser treatment it was not included in the study. However, it is an area of interest and some confusion, so the following section has been added to the Discussion, Lines 380-416:

Photobiomodulation

While not properly laser treatment, photobiomodulation (PBM) has recently received attention for treatment of dry AMD and bears comment. 46 Popular for many years, especially in Eastern Europe, use of visible light exposure, commonly referred to as “red light therapy”, has recently been approved by the United States Food and Drug Administration (FDA) for treatment of dry AMD. 47 The simplest distinction between retinal laser treatment and PBM is that retinal laser is thermal, while PBM is not. Unlike retinal laser treatment whose therapeutic effects are mediated by laser-induced retinalhyperthermia causing low-dose adaptive hormesis in the retina initiated by RPE HSP activation (the catabolic phase of the hormetic response) resulting in normalized proteostasis and thus normalized cell function (the anabolic phase of the hormetic response), reduced inflammation, and therapeutic immunoactivation, PBM operates via an entirely different mechanism and thus has different and generally a more narrow range of effects. 3, 6, 23,27, 45-49  In PBM, photons from visible light, generally produced by incandescent bulbs or light-emitting diode sources, are absorbed by electrons in the metal cations of the respiratory chain molecules, energetically promoting them temporarily into a higher energy valence. Just as quickly the electron orbit degrades to return to its native state releasing the absorbed energy into the system. This is termed the “photoelectric effect”. 3, 48 The principal result is to temporarily improve ATP production and mitochondrial function. The effects of non-thermal PBM are highly wavelength dependent and order-of-presentation specific, while the thermal effects of retinal laser are not. 3,46-48 Because of the different mechanisms of action, the effects of retinal laser tend to be more therapeutically diverse, longer acting, and more robust. 3, 49

In the Lightsite III trial examining use of the Valeda PBM device (Valeda Light Delivery System (LumiThera, Inc., Poulsbo, WA), compared Valeda PBM treated eyes with “sham” treatment (lower intensity illumination) over 13 months. 47 100 patients (148 eyes) eyes were eligible for randomization. 91 treated and 54 sham eyes entered the study, however exceeding the planned drop-out rate of 10%, 31% of randomized patients failed to complete study that required an onerous 9 treatment sessions over 3-5 weeks every 4 months, resulting in just 79 treated and 40 control eyes completing the study. FDA approval appeared to rely on a modest 2.4 letter average visual acuity improvement in treated patients (p=0.02) over controls at the end of 13 months. Secondary outcomes including low-light best corrected VA, Radner reading chart performance, and Visual Quality of Life (VFQ) – 25 scores were unchanged in both treated and control eyes. The development of new GA appeared reduced in treated eyes (p=.0024) compared to controls. However, while the authors describe a “favorable safety profile”, three times as many treated eyes developed neovascular AMD than controls (5.4 vs 1.8%), a serious adverse treatment effect suggesting the possibility of phototoxicity. 47 Thus, despite regulatory approval, questions remain and concerns have been expressed regarding the Lightsite III trial and the efficacy and safety of PBM for dry AMD. 50

Introduction

General: I have not seen you introduce the abbreviation SDM. 

We note that SDM is defined in lines 223-224 of the text

Line 45: there is no therapy without possible side effects, thus “can be employed as a medical therapy without adverse effects” seems somewhat of an exaggeration.

SDM is indeed fairly unique in that no adverse treatment effects of any kind have ever been identified in over 25 years of clinical and experimental / laboratory treatment. This has been very well established.

Results/Discussion

Line 63: dating back several millennia – including the treatment of the eye? Is that really true? Several millennia would imply some 3000+years back. Please reword if the millennia only concern other applications of lasers.

This has been reworded to state: “…a millenium or more….”

Round 2

Reviewer 2 Report

Comments and Suggestions for Authors

The requested comments are addressed. No further comment.